# The Added Value of Atrial Strain Assessment in Clinical Practice

**DOI:** 10.3390/diagnostics12040982

**Published:** 2022-04-13

**Authors:** Andrea Ágnes Molnár, Béla Merkely

**Affiliations:** Heart and Vascular Center, Semmelweis University, 1122 Budapest, Hungary; merkely.bela@gmail.com

**Keywords:** speckle tracking, strain, atrial function, predictive value

## Abstract

Speckle tracking echocardiography has emerged as a sensitive tool to analyze myocardial function with improved diagnostic accuracy and prognostic value. Left atrial strain assessment has become a novel imaging method in cardiology with superior prognostic value compared to conventional left atrial volume indices. Left atrial function is divided into three phases, reservoir function being the most important. This review summarizes the added value of speckle tracking echocardiography derived left atrial strain assessment in clinical practice. Recently published data suggest the prognostic value of left atrial reservoir function in heart failure, atrial fibrillation, stroke and valvular heart disease. Furthermore, left atrial reservoir strain proved to be a predictor of cardiovascular morbidity and mortality in the general population. Thus, routine assessment of left atrial function can be an optimal strategy to improve cardiovascular risk prediction and supplement the current risk prediction models.

## 1. Introduction

Two-dimensional speckle tracking echocardiography (2DSTE) has become an increasingly used imaging method in clinical practice mainly due to its high sensitivity to evaluate left ventricular (LV) systolic function. Left ventricular global longitudinal strain (LV GLS) has been proven to predict cardiovascular mortality and morbidity [1]. Recently, left atrial (LA) function assessment has gained increasing attention using the same 2DSTE method [2]. Left atrial function is divided into three phases: reservoir, conduit and contraction function. Reservoir strain (also referred as peak atrial longitudinal strain—PALS) is considered the most important due to its prognostic value in cardiovascular diseases [3]. Moreover, LA reservoir strain has been confirmed as predictor of cardiovascular morbidity and mortality in the general population [4]. Cardiovascular diseases compromising pressure and volume overload or arrhythmic insult can result in adverse atrial myocardial tissue remodeling that increases atrial interstitial fibrosis. This leads to a decrease in atrial elasticity resulting in impaired compliance, which was most accurately detected by reservoir LA strain compared to all echocardiographic parameters [5]. Thus, the amount of LA interstitial fibrosis determines LA reservoir function [5]. Advances in cardiac imaging in the field of speckle tracking echocardiography, including machine learning algorithms, provide a more reliable, fast and feasible on-site examination method of the left atrium. Thus, speckle tracking echocardiography has become an increasingly standard imaging method in routine clinical practice. Left atrial function can be an optimal strategy to improve cardiovascular risk prediction and supplement the current risk prediction models [4,6].

## 2. Speckle Tracking Echocardiography Method to Assess Left Atrial Function

Left atrial volumes and functions are assessed using either 2DSTE or tissue Doppler imaging. However, 2DSTE is less angle- and load-dependent compared to tissue Doppler imaging that tracks the image acoustic markers to evaluate atrial strain. Atrial strain is the fractional change in the length of the entire atrial myocardium contour in the tangential direction referred as atrial longitudinal strain [7]. Strain rate represents the speed at which myocardial deformation occurs [7]. The LA longitudinal strain curve is composed of three phases according to the atrial physiology: reservoir, conduit and contraction phase [7] (Figure 1). The reservoir phase features LA reservoir function during systole, when pulmonary veins fill the left atrium leading to stretched atrial wall, and corresponds to LV isovolumic contraction, ejection and isovolumic relaxation. LA conduit phase characterizes LA conduit function during early diastole when the mitral valve opens and the left atrium empties to the LV which corresponds to the E wave (early transmitral flow) that is modulated by atrial compliance, LV relaxation and compliance. The last phase represents LA contraction and the LA pump function, corresponding to the A wave (late transmitral flow) which depends on atrial preload (venous return) and atrial afterload (LV end-diastolic pressure) [7,8]. All phases of LA function are modulated by both loading conditions and heart rate.

The EACVI/ASE/Industry Taskforce consensus document recommended standardized LA strain analysis using apical four chamber non-foreshortened views with higher frame rate (50–70 frames/min) [7]. The use of both apical four and two chamber views is optional [7] (Figure 2). Zero-baseline for LA strain curve is recommended to be set at ventricular end-diastole using R-R ECG gating; however, a reference set at the onset of LA contraction using P-P gating is also accepted [7]. Nevertheless, the two methods result in different strain values, as the LA length is different at these two zero points. Thus, all LA strain absolute values are higher in case of R-R gating compared to P-P gating [7]. ECG trigger R-R gating, as time reference for end-diastole, can be misleading in several ECG alterations such as bundle branch block [7]. In this case, end-diastole can be determined according to the nadir of the LA strain curve. Furthermore, P-P gating cannot be used in case of atrial fibrillation (AF) [7]. The value of peak atrial longitudinal strain (PALS, also referred as peak reservoir strain) does not overlap in all cases with the longitudinal strain value of ventricular end-systole. Therefore, distinct labeling is suggested in such cases, e.g., LA peak reservoir strain or LA end-systolic reservoir strain. Nevertheless, reservoir LA function assessed by 2DSTE compromises only the longitudinal LA strain and neglects the radial and transverse strain components because of technical reasons. Moreover, orifices of pulmonary veins and LA appendage need tracking extrapolation; thus, sub-division of the LA wall into segments is not recommended, and LA longitudinal strain is referred to as the global longitudinal strain (LA GLS) of the entire wall [7,8]. Further methodological challenges may occur when contouring the LA endocardial surface due to the thin LA wall and mobile interatrial septum [8]. Interatrial septal aneurysms may represent another issue during endocardial tracking (Figure 3a,b). Besides, the far field localization of LA on the transthoracic image worsens lateral resolution [8]. However, speckle tracking echocardiography of the left atrium becomes an increasingly standard, feasible and fast imaging method due to advances in cardiac imaging including machine learning algorithms.

## 3. Left Atrial Strain in Healthy Individuals

The normal reference value for LA reservoir strain was assessed by many working groups in the last two decades. Overall, it can be concluded that the normal value of LA reservoir strain varied across publications depending on sample size, vendor, ECG gating, age, gender and racial differences. Meel et al. revealed reservoir strain of 39% in a healthy black population of 120 individuals; meanwhile, Cameli and coworkers found 42% as a reference value examining 60 healthy subjects in Italy [9,10]. Sun et al. reported a somewhat higher normal LA reservoir strain value of 47% in a healthy cohort of 121 individuals from Hong Kong [11]. The lowest normal reference reservoir strain value (23%) was reported by Saraiava et al. in 64 individuals owing to the different measurement method using P-P ECG gating [12]. Most of the studies used R-R ECG gating in the literature, when the atrial strain values are higher compared to P-P ECG gating values. A recently published meta-analysis of 2542 healthy individuals showed that normal value for reservoir strain is 39%, while conduit strain is 23% and contractile strain is 17% [13]. In a large cohort of 2812 healthy Asian individuals, Liao and coworkers showed that LA reservoir strain decreases with aging and high blood pressure, which is compensated by increasing atrial contraction strain [14]. The Normal Reference Ranges for Echocardiography (NORRE) multicenter study found that both LA reservoir and conduit function decrease with age, while pump function increases in healthy adults [15]. In a study by van Grootel, the healthy subjects were divided into five age decades. Van Grootel and coworkers demonstrated that LA reservoir strain and early diastolic LA strain rate are lowest in the older population, while late diastolic LA strain rate is higher [16]. Most of the studies reported that gender has no effect on LA strain values in healthy cohorts [13,15,17]. On the contrary, Liao and coworkers found that women have higher LA reservoir function at younger ages compared to men (39.34 ± 7.99% vs. 37.95 ± 7.96%), but the age-dependent LA reservoir strain deterioration is more pronounced in women [14]. In a cohort of 329 healthy subjects, Morris and coworkers presented that the normal value of PALS ranged between 23% and 45% [17]. Furthermore, when analyzing race variations in LA function, there were no significant differences between Asian and European age-matched healthy subjects [17]. The validation group in the study of Morris and coworkers comprises a further 377 patients with LV diastolic dysfunction [17]. The LA strain in this validation group was negatively correlated with New York Heart Association functional classification stadium [17]. Cauwenberghs and coworkers demonstrated that subclinical LA dysfunction, defined as PALS lower than 23%, was associated with a 111% increased higher adjusted risk for future adverse cardiac events in the general population [18].

## 4. Left Atrial Strain Assessment in the Presence of Cardiovascular Risk Factors

Hypertension, diabetes, obesity and sleep apnea syndrome are known risk factors to promote LA remodeling [19,20]. LA dysfunction and stiffness were more pronounced in hypertensive patients compared to healthy subjects when LV hypertrophy was present [21]. Hyperglycemia can induce an elevation of pro-fibrotic signaling molecule level that provokes collagen synthesis and interstitial fibrosis [22]. Fibrosis decreases the elasticity of the LA wall and may provoke AF. Previous publications reported decreased atrial function in diabetic and pre-diabetic populations using the speckle tracking method [23,24,25]. However, LA dysfunction could not be revealed when using volumetric method for assessment of LA phasic function in diabetic and hypertensive population [23]. Mondillo and coworkers demonstrated gradual reduction in LA strain parameters from hypertensive and diabetic patients to patients with concomitant hypertension and diabetes [23]. Similarly, concomitant obesity and diabetes significantly worsen LA function when assessing them by strain analysis; however, volume-derived LA functional analysis did not confirm these results. [26]. In cases of coronary artery disease with poorly controlled diabetes, impaired LA conduit function was found using the strain analysis method [27]. Furthermore, the Multi-Ethnic Study of Atherosclerosis (MESA) revealed that LA function was predictive for incidence of cardiovascular disease in a large-scale, asymptomatic diabetic patient population, as measured by emptying fraction using a sensitive tissue tracking cardiac magnetic resonance imaging method [28]. Overall, these results support the hypothesis that LA strain analysis might hold high sensitivity to detect more profound cardiac changes [25].

It is well known that chronic kidney disease represents increased risk of cardiovascular mortality and morbidity [29]. However, calculation of the Framingham risk score underestimates cardiac events in patients with chronic kidney disease. Chronic kidney disease may result in cardiac dysfunction independently of the presence of hypertension and diabetes [30]. Previous publications demonstrated cardiac alterations in patients with advanced stages of chronic kidney disease using conventional echocardiographic measurements [31]. The increased activation of the renin–angiotensin–aldosterone system in chronic renal failure may lead to myocardial fibrosis and consequent LA remodeling [30]. Kidney disease also leads to fluid overload, which in turn may result in LA dilatation. Coexistent hypertension, diabetes or coronary artery disease may also contribute to LA remodeling in chronic kidney disease population. Tripepi and coworkers found that LA volume alterations predict cardiovascular events in patients with end-stage renal failure and dialysis treatment [32]. Moreover, in early stages of chronic renal disease, Kadappu and coworkers showed that LA strain has incremental predictive value over conventional echocardiographic parameters to detect myocardial involvement [30].

The Copenhagen City Heart Study was designed as a longitudinal cohort study to identify cardiovascular risk factors in a general population comprising 385 participants [4]. The mean LA reservoir strain in the Copenhagen City Heart Study was 37.4%, which was significantly associated with LV GLS and the ratio between early mitral inflow velocity and mitral annular early diastolic velocity (E/e′) [4]. Diastolic function and LV filling pressure determines left atrial volume, which determines LA reservoir strain [33]. In the Copenhagen City Heart Study, LA reservoir strain proved to be an independent predictor of cardiovascular morbidity and mortality only in women, and its prognostic value was incremental to predictors from SCORE and the ACC/AHA Pooled Cohort Equation [4].

## 5. The Added Value of LA Strain Evaluation in Cardiac Diseases

### 5.1. Usefulness of LA Strain Assessment in Heart Failure

The prevalence of heart failure is increasing worldwide [34]. Almost 50% of all heart failure cases are due to heart failure with preserved and mildly reduced ejection fractions (HFpEF and HFmEF) [34]. Diastolic dysfunction represents the main pathomechanism of HFpEF. The echocardiographic assessment algorithm of diastolic function is complex and multiparametric, using transmitral flow, tissue velocity, maximum left atrial volume index (LAVi) and estimated pulmonary artery pressure [34,35,36]. LAVi and E/e’ initially rise with worsening of diastolic dysfunction but both lose their diagnostic value at more severe grades of diastolic dysfunction (e.g., when separating grade II diastolic dysfunction from grade III) [37]. Increased LV filling pressure dilates LA chamber and stretches atrial cardiomyocytes; thus, consequently releasing natriuretic peptide secretion and leading to LA dysfunction [38]. Santos and coworkers demonstrated that LA functional parameters are lower in HFpEF patients compared to healthy controls, independently from LA size [39]. Moreover, LA function assessment provides additive value in diagnosing, grading severity and monitoring treatment effect in diastolic dysfunction [3]. As LA strain shows a gradual progression between all stages of diastolic dysfunction, it represents a sensitive diagnostic tool to differentiate between stages of advanced diastolic dysfunction [37]. Singh and coworkers found that an LA strain cutoff of 19% separates grade III diastolic dysfunction with >90% accuracy, and an LA strain threshold of >24% differentiates grade I from grade II or grade III diastolic dysfunction [37]. Results from the Karolinska–Umeå (KARUM) hemodynamic database demonstrated that left atrial reservoir strain improves diagnostic accuracy of the 2016 ASE/EACVI diastolic algorithm in patients with HFpEF [40].

Heart failure with reduced EF (HFrEF) is defined when symptoms and/or signs of heart failure are present and the LV systolic function is reduced (LV EF ≤40%) [34]. It is well known that LA dilatation is associated with higher mortality rate and the coexistence of LA dilatation and dysfunction is frequent [41]. However, the overlap between LA dilatation and functional impairment is variable [42]. LA reservoir strain emerged as a significant prognostic parameter in HFrEF independently of LA volume, LV GLS, age, LV EF and E/E′ ration [43]. Increased LA stiffness is characteristic in HFrEF and is an important determinant of pulmonary hypertension. Cardiac resynchronization therapy (CRT) is a well-established treatment for selected patients with heart failure. The resynchronization of the heart may lead to LV reverse remodeling [34]. Consequently, CRT may reduce the grade of functional mitral regurgitation and may improve diastolic function in CRT responder population [34]. Furthermore, LV reverse remodeling can induce LA reverse remodeling as well [44]. LA reservoir strain increase after CRT is associated with favorable long-term outcomes [45]. Stassen and coworkers found that patients with both LV and LA reverse remodeling after CRT implantation have better outcomes compared to patients with only LV or LA reverse remodeling or without any reverse remodeling [44]. The combined LA and LV GLS assessment has incremental prognostic value over previously established CRT responder definition parameters including >15% reduction in LV end-systolic volume [44]. Overall, the usefulness of LA strain measurement seems somewhat greater in HFpEF than in HFrEF.

### 5.2. The Added Value of Atrial Strain Evaluation in Valvular Heart Disease

Aortic valve stenosis (AS) and mitral regurgitation (MR) are among the most prevalent valvular heart diseases in the Western world [46]. The consequence of aortic stenosis is LV pressure overload, which leads to LV hypertrophy and myocardial perfusion abnormalities, interstitial fibrosis and diastolic dysfunction with increased LV filling pressures. The left atrium dilates and the functional parameters of the left atrium decrease, resulting in arrhythmias [47]. In cases of severe AS, both LA volumes and LA strains are independent predictors of cardiovascular events; however, LA dysfunction may present earlier than LA dilatation [17,48,49]. During the last decades, a large number of studies paid attention to the cardiovascular risks and progression of moderate AS and asymptomatic severe AS [50,51,52]. Patients with moderate AS have an increased risk of mortality compared to the general population [52]. The main prognostic indices in asymptomatic and moderate AS were LV mass index, pulmonary hypertension, diabetes mellitus, rapid increase in aortic-jet velocity, value of plasma N-terminal pro–B type natriuretic peptide (NT-proBNP), average E/e’, LAVi and LA GLS [50,51,52,53,54]. Moreover, LA GLS revealed incremental prognostic value over LAVi [50]. Aortic valve replacement represents the treatment for severe AS [55]. According to the current guidelines, surgical aortic valve replacement (SAVR) is indicated in younger patients with lower surgical risk, while transcatheter aortic valve implantation (TAVI) is recommended in older patients above 75 years and/or in patients with high surgical risk [55]. New-onset AF occurs 40% after valve surgery, representing increased risk of morbidity and mortality [56,57]. Cameli and coworkers found that a cut-off value of 16.9% for preoperative LA reservoir strain accurately predicts atrial dysfunction and the risk of AF after SAVR [58]. Reverse LA remodeling with a mild improvement in LA reservoir function could be detected after TAVI [59]. Furthermore, LA GLS, LA volumes, and LV GLS were associated with post-implantation adverse clinical outcomes after TAVI [59].

Chronic primary (organic) and secondary (functional) mitral regurgitation (MR) induces LA structural and functional remodeling, including interstitial fibrosis due to volume overload, leading to increased vulnerability to AF [60,61,62]. Consequently, mitral valve disease represents a major risk factor for AF [60,61,62]. Cameli and coworkers demonstrated that LA reservoir strain inversely correlates with the degree of MR; thus, it is lower in severe MR compared to mild MR with an additional reduction in patients with paroxysmal AF [60,63]. Surgery is recommended in symptomatic patients with low surgical risk [55]. Furthermore, current guidelines recommend surgery even in asymptomatic patients with primary MR if LV EF decreases below 60%. In case of high surgical risk, transcatheter edge-to-edge mitral valve repair should be considered in symptomatic patients with secondary mitral regurgitation eligible for intervention [55] (Figure 2). Both surgical or percutaneous mitral valve interventions reduce preload, LA volumes and LA extension due to a reduction in mitral regurgitation [64,65]. Controversial findings have been reported so far regarding LA volumetric and functional alterations after transcatheter edge-to-edge mitral valve repair using MitraClip implantation [66,67,68]. Rammos et al. found conduit and contraction LA function improvement in patients with functional MR after MitraClip procedure [67]. Similarly, Toprak et al. found improved LA strain rates after MitraClip implantation in a small-scale study compromising mainly patients with functional MR [68]. On the contrary, Öztürk and coworkers found no relevant changes during follow-up after MitraClip implantation; however, the baseline reservoir strain was the strongest predictor for mortality and adverse outcomes. [66]. In the retrospective study of Avenatti and coworkers, LA stiffness, defined as the ratio of change in LA pressure to change in LA volume during systole, improved after MitraClip implantation irrespective of the etiology of MR [69]. In the EVEREST II trial (Endovascular Valve Edge-to-Edge Repair Study II), Ipek and coworkers found that LA strain values after mitral valve repair (transcatheter or surgical) depend on baseline strain values [70]. Reduced baseline LA strain values remained reduced after repair despite significant MR and LA volume reduction [70]. Candan and coworkers found that patients who developed postoperative AF after mitral valve surgery had preexistent LA dysfunction [71]. The preexistent LA dysfunction, when superimposed by the acute stress of surgical intervention, manifested clinically as postoperative AF. [71]. Preoperative LA reservoir strain and LAVi were independent predictors of new-onset AF after surgery for mitral valve regurgitation [71]. Furthermore, debate questions still remain for asymptomatic patients with moderate MR regarding optimal time for surgery. Atrial strain analysis might help to identify subgroups of these patients who would benefit from intervention. Cameli and coworkers showed that LA reservoir strain with a cut-off value of <35% proved to be a useful marker of cardiovascular events in asymptomatic patients with moderate MR [72].

### 5.3. LA Strain Represents a New Sensitive Parameter in Cancer-Therapy-Related Cardiac Dysfunction

Early detection of chemotherapy-related cardiomyopathy is crucial to minimize cancer-treatment-induced heart disease. Transthoracic echocardiography is the gold standard method to evaluate patients during and after chemotherapy. Besides conventional functional parameters, current guidelines recommend speckle-tracking-derived LV GLS assessment due to its sensitive and predictive value to identify subclinical LV deterioration [73]. Cancer-therapy-related cardiac dysfunction (CTRCD) is defined as a reduction in LVEF below 53% or a 10% reduction from baseline to below the lower normal limit [73]. Furthermore, changes in LV GLS may be considered as an early sign of CTRCD [73]. Recently, research focus turned from the systolic LV function changes to LA functional alterations in cancer-treated populations. Laufer-Perl and coworkers found that 50% of patients treated with anthracycline showed a 10% relative reduction in LA reservoir strain and/or a decrease in LA reservoir strain below 35% [74]. Park and coworkers found that both LA reservoir strain and LV GLS deteriorated before developing overt CTRCD; however, LA reservoir strain reduction showed better sensitivity and specificity in predicting CTRCD than LV GLS [75]. Furthermore, Laufer-Perl M. and coworkers found that even before chemotherapy, LA reservoir strain was lower in 17% of cancer patients [74]. Similarly, Tadic and coworkers showed that LA reservoir and conduit function was reduced, while LA contraction function was elevated in the cancer group, which might represent a compensatory mechanism to maintain LV filling [76]. Nonetheless, it is debatable whether LV diastolic dysfunction is only the consequence of anti-cancer therapy or if cancer itself leads to it. Interestingly, the correlation between cancer and LA dysfunction was independent of common risk factors for LA remodeling such as age, obesity, hypertension, diabetes, LV hypertrophy and LV diastolic function [76]. Thus, both cancer treatment and cancer disease itself can lead to LA functional impairment, which is associated with a higher risk of AF and which represents a frequent arrhythmia in cancer populations [77,78]. Radiotherapy, including the heart in the radiation field, induces late manifestations of cardiac damage in a dose-dependent manner, involving valvular heart disease, constrictive pericarditis, cardiomyopathy, coronary artery disease and arrhythmia [73]. Early identification of patients at higher cardiotoxic risk might improve long-term outcomes; however, there are scarce data regarding the utility of strain imaging to investigate radiotherapy-induced myocardial damage [76,79,80].

### 5.4. Predictive Value of LA Strain in Atrial Fibrillation and Stroke

Atrial fibrillation is the most common cardiac arrhythmia with increasing prevalence leading to heart failure and peripheral embolism [81,82]. Asymptomatic (silent) paroxysmal AF occurs in at least 30% of cryptogenic stroke population, representing high diagnostic and clinical burden, as the first manifestation of arrhythmia in many cases is systematic embolism [83]. Moreover, early detection of silent paroxysmal AF in cryptogenic stroke is still challenging, resulting in 5% of annual stroke recurrence [84]. The sensitivity of 24 h ECG monitoring with Holter device is limited for the detection of new-onset AF [85]. Current guidelines recommend 30 days of ECG monitoring within 6 months of a cryptogenic stroke (Class II.a.; level of evidence C) [86]. The Cryptogenic Stroke and Underlying Atrial Fibrillation (CRYSTAL AF) randomized study showed that ECG monitoring with an insertable cardiac monitor (ICM) was superior to conventional follow-up for detecting AF after cryptogenic stroke [87]. At 6 months, AF was detected in 1.4% and 8.9% in the conventional follow-up and ICM group, respectively [87]. Moreover, the detection of AF at 36 months after cryptogenic stroke was 30% in the ICM group and 3% in the conventional follow-up group [88]. Nonetheless, ICM is an invasive and not widely applicable method to detect AF; thus, there is an unmet need to improve our current risk stratification scores with the implementation of further sensitive and predictive clinical parameters to identity potential patients at higher risk of AF and stroke.

Left atrial structural, functional and electrical remodeling is a common underlying alteration in AF [89]. Atrial stunning is a frequent phenomenon after spontaneous or electrical and pharmacological cardioversion, indicating that it is a consequence of the underlying arrhythmia and possible atrial myopathy [90]. Left atrial strain assessment using speckle tracking echocardiography to evaluate LA function may help to predict the risk of AF and stroke [91,92,93]. The SURPRISE (Stroke Prior to Diagnosis of Atrial Fibrillation Using Long Term Observation with Implantable Cardiac Monitoring Apparatus Reveal) echo sub-study was designed to assess echocardiographic predictors of paroxysmal AF in cryptogenic stroke patients who had ICM for 3 years [91]. LA reservoir strain was associated with paroxysmal AF, and this was independent of LV GLS and LA size [91]. Besides, in the meta-analysis of Sachdeva et al., reduced LA GLS was predictive of cerebrovascular stroke in AF patients [92]. Furthermore, Pathan and coworkers showed incremental value of LA strain above current risk-prediction models for predicting AF in cryptogenic stroke [93]. Catheter ablation therapy has become a cornerstone in the treatment of symptomatic paroxysmal or persistent AF [94]. Previous publications demonstrated that LA GLS independently predicted AF recurrence after ablation with a cut-off value of 17–18% [95,96,97]. Left atrial reservoir strain is a marker of LA fibrosis representing an underlying substrate predisposing for AF, while atrial contraction function is as a marker of atrial stunning. The use of atrial functional parameters in cryptogenic stroke populations can help to identify patients who may benefit from long-term rhythm monitoring and potential anticoagulation therapy [98].

Left atrial strain assessment may help to identify patients at risk of atrial fibrillation in cases of cardiomyopathies such as hypertrophic cardiomyopathy and dilative cardiomyopathy [99,100]. Atrial strain values were significantly lower in dilative cardiomyopathy compared to ischemic cardiomyopathy presumably due to the higher degree of atrial fibrosis in dilative cardiomyopathy [101,102]. Atrial arrhythmias are common in patients with restrictive cardiomyopathy, particularly in cardiac amyloidosis, representing a higher risk population for thromboembolism [103,104]. In a large cohort of patients with cardiac amyloidosis, Bandera and coworkers found increased atrial stiffness and reduced left atrial reservoir and contraction function, which was independently associated with reduced clinical outcomes [103]. Atrial dysfunction in this population was not attributable only to the consequence of LV dysfunction, but also to the amyloid deposition in the atrial wall [103]. Furthermore, Bandera and coworkers revealed that one fifth of patients with sinus rhythm and cardiac amyloidosis showed no evidence of atrial contraction when using LA strain analysis [103]. Moreover, atrial electromechanical dissociation was associated with a worse prognosis compared to patients in sinus rhythm with mechanical atrial contraction [103]. Consequently, atrial strain analysis showed additive value to identify higher thromboembolic risk patients with cardiac amyloidosis and sinus rhythm compared to conventional clinical workup due to its strength in detecting atrial electromechanical dissociation [103]. However, the indication of anticoagulation in this population is still unsettled [104].

## 6. Characteristics of Left Atrium in “Athlete’s Heart”

Extensive exercise training leads to physiological adaptation of the heart including left atrial structural and functional remodeling [105,106]. There are scarce data regarding the physiologic alterations of left atrium in “athlete’s heart”. Previous publications found that both LA reservoir and contractile functions are lower-normal, while LA volumes are higher in athletes [107,108]. These alterations are influenced by age, gender, ethnicity, type of sport, cumulative years of training and degree of conditioning. The atrial enlargement and the low-normal function in athletes may not denote dysfunction but a physiologic aspect of the ‘‘athlete’s heart’’. Less powerful contraction of an enlarged left atrium can result in the same atrial stroke volume as a more robust contraction of a smaller atrium [106,108]. However, the prevalence of AF is significantly higher in athletes, which is supposed to be associated with left atrial remodeling [109]. Comparing athletes and non-athletes with and without AF, both athletes and non-athletes with AF had reduced atrial strain. However, in the non-athlete group with AF, additional diastolic dysfunction was observed. In the athlete group with AF, only left atrial enlargement was pronounced without diastolic dysfunction [109]. Thus, the pathomechanism of atrial myopathy and AF might be different in athletes and non-athletes. In athletes, exercise-induced hemodynamic atrial stretch may be the main risk factor, while in non-athletes, diastolic-dysfunction-induced atrial stretch might be the main trigger [109].

## 7. LA Strain in COVID-19 Pneumonia-Associated Atrial Fibrillation

Coronavirus disease 2019 (COVID-19) is caused by the severe acute respiratory syndrome coronavirus 2 (SARS-CoV-2) RNA virus, which has affected the health of millions worldwide in the last years [110]. Severe complications developed in 15% of COVID-19 patients, such as acute respiratory distress syndrome, acute myocardial injury or arrhythmia [111]. Among patients with cardiac disease and COVID-19 pneumonia, atrial fibrillation was the most common documented arrhythmia with an incidence up to 36% [112]. Hypoxemia, systemic inflammation, alterations of the renin-angiotensin system and a higher presence of cardiac comorbidities are the main pathomechanisms to develop new-onset AF in severe COVID-19 disease [113]. New-onset AF is associated with increased mortality in critically ill patients and is strongly associated with mechanical ventilation, organ failure and norepinephrine use [114,115]. Thus, prediction of AF risk is important. In the study of Beyls and coworkers, 20% of patients admitted to the Intensive Care Unit because of COVID-19 pneumonia developed new-onset AF [115]. LA reservoir and conduit strain values were significantly impaired in the AF group with COVID-19 pneumonia; however, only LA conduit strain remained independently associated with AF [115]. Furthermore, Beyls and coworkers reported a cut-off value of −11% for LA conduit strain to predict new-onset AF in COVID-19 pneumonia with a sensitivity of 76% and a specificity of 75% [115].

## 8. Conclusions

Previously, LA examination was limited to the evaluation of LA size in the clinical practice. The on-site assessment of LA function during routine echocardiography examination was neglected due to the lack of dedicated software and standardized methodology. However, there is growing evidence that LA remodeling is not equal to LA enlargement alone as it encompasses LA changes at the molecular, cellular and tissue level even before LA enlargement [116]. Advances in cardiac imaging, including machine learning algorithms, can provide a more feasible on-site examination of LA function using speckle tracking echocardiography in clinical routine. LA function has been assessed in several clinical settings, concluding that LA reservoir strain is a sensitive prognostic marker in many cardiac diseases involving hypertension, diabetes, chronic renal failure, chronic heart failure, valvular heart disease, AF and cardiotoxicity [19,26,30,50,59,76,91,92,93]. The first manifestation of asymptomatic AF in many cases is systematic embolism; thus, early diagnosis of arrhythmia is of utmost clinical importance. LA reservoir strain is a marker of atrial fibrosis representing an underlying substrate predisposing for AF [5,91]. Introducing LA strain measurement in the routine echocardiographic examination might help to evaluate subclinical LA dysfunction and risk stratification for the occurrence of AF. Furthermore, left atrial reservoir strain proved to be a predictor of cardiovascular morbidity and mortality in the general population [4]. Atrial enlargement and the low-normal function in athletes may not denote dysfunction but a physiologic cardiac adaptation to extensive exercise training. In elite athletes, however, extensive atrial enlargement without diastolic dysfunction might be the main pathomechanism of atrial myopathy leading to AF [106]. In conclusion, adding LA reservoir strain to the routine echocardiographic report may revise the current disease grading and risk scores and may help to detect subtle cardiac dysfunction.

## Figures and Tables

**Figure 1 diagnostics-12-00982-f001:**
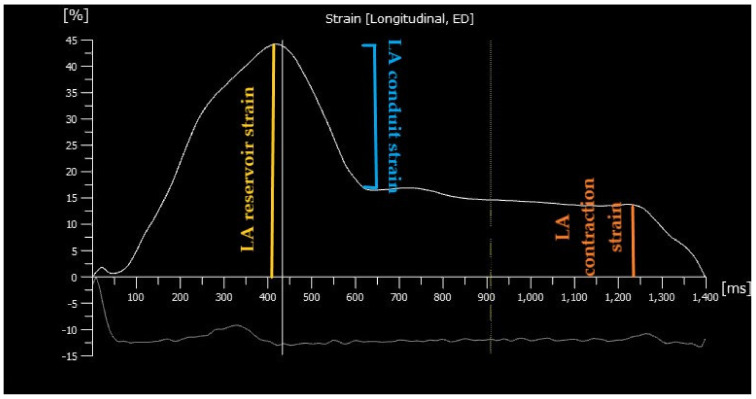
Speckle tracking analysis and strain measurements during the three phases of left atrial (LA) cycle using R-R ECG gating with the zero-baseline strain reference set at ventricular end-diastole (ED). Reservoir strain is measured as the difference between the peak strain curve value and baseline (ED) strain value (positive value). Conduit strain is calculated as difference of the strain value at the onset of atrial contraction minus the peak strain value (negative value). Contraction strain is calculated as difference of the strain value at baseline (ED) minus the strain value at onset of atrial contraction (negative value). The contraction phase is missing in case of atrial fibrillation.

**Figure 2 diagnostics-12-00982-f002:**
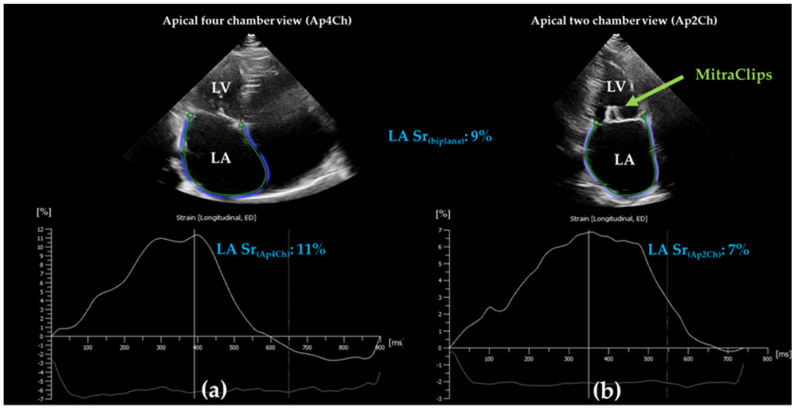
Left atrial speckle tracking analysis shows decreased strain values using apical four- (**a**) and two-chamber (**b**) views three months after MitraClip implantation (2 clips, green arrow). Apical four-chamber view is recommended to assess left atrial strain. However, the use of both apical four- and two-chamber views, together with the average of the strain values, is optional. Panel (**a**) and (**b**) demonstrate the difference of reservoir strain values between the two methods. The contraction strain is missing from the strain curves because of atrial fibrillation. LASr: reservoir strain; LA: left atrium; LV: left ventricle; ED: end-diastole.

**Figure 3 diagnostics-12-00982-f003:**
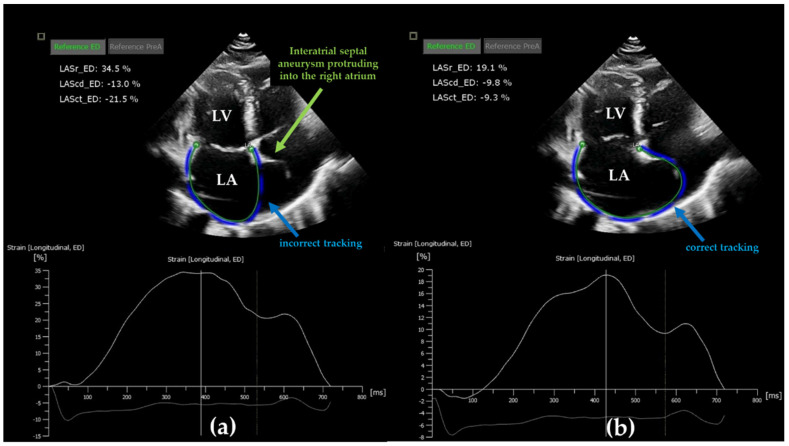
Methodological challenges of left atrial strain assessment in case of interatrial septal aneurysm. (**a**) Incorrect automated delineation of endocardial surface in case of interatrial septal aneurysm. The automated delineation neglects the true contour of interatrial septal aneurysm protruding into the right atrium. (**b**) Manual correction of endocardial tracking results in different strain values compared to the measurements of panel (**a**), both performed in the same patient. LASr: reservoir strain; LAScd: conduit strain; LASct: contraction strain; LA: left atrium; LV: left ventricle; ED: end-diastole.

## Data Availability

Not applicable.

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
