# Peer review of "The Added Value of Atrial Strain Assessment in Clinical Practice"

_diagnostics, 2022, doi:10.3390/diagnostics12040982_

Round 1
Reviewer 1 Report
This review is well written; it analyzes very clearly the various applications of left atrial strain.
The authors should add informations about the role of left atrial strain in cardiomyopathies and in particular in cardiac amyloidosis. The authors could cite a recent study of Bandera (doi: 10.1016/j.jcmg.2021.06.022) on atrial strain in cardiac amyloidosis and a recent review published in Current Problems in Cardiology (doi: 10.1016 / j.cpcardiol.2022.101188) on anticoagulation in cardiac amyloidosis in the presence of electromechanical atrial dissociation.
Author Response
Response to Reviewer #1
We would like to express our thanks to Reviewer#1 for the careful evaluation of the manuscript and the helpful and constructive suggestions. The Reviewer recommended adding information about the role of left atrial strain in cardiomyopathies and in cardiac amyloidosis. Furthermore, to supplement the Reference list with a recent study of Bandera (doi: 10.1016/j.jcmg.2021.06.022) on atrial strain in cardiac amyloidosis and a recent review published in Current Problems in Cardiology (doi: 10.1016 / j.cpcardiol.2022.101188) on anticoagulation in cardiac amyloidosis in the presence of electromechanical atrial dissociation. According to the Reviewer’s suggestions, we supplemented the manuscript text and the Reference list with the following content highlighted in yellow:
Manuscript text - page number 9:
“Left atrial strain assessment may help to identify patients at risk of atrial fibrillation in case of cardiomyopathies such hypertrophic cardiomyopathy and dilative cardiomyopathy [101,102]. Atrial strain values were significantly lower in dilative cardiomyopathy compared to ischemic cardiomyopathy presumably due to the higher degree of atrial fibrosis in dilative cardiomyopathy [103,104]. Atrial arrhythmias are common in patients with restrictive cardiomyopathy, particularly in cardiac amyloidosis, representing a higher risk population for thromboembolism [105,106]. In a large cohort of patients with cardiac amyloidosis, Bandera and coworkers found increased atrial stiffness and reduced left atrial reservoir and contraction function, which was independently associated with reduced clinical outcomes [105]. Atrial dysfunction in this population was not attributable only to the consequence of LV dysfunction, but also to the amyloid deposition in the atrial wall [105]. Furthermore, Bandera and coworkers revealed, that one-fifth of patients with sinus rhythm and cardiac amyloidosis showed no evidence of atrial contraction when using LA strain analysis [105]. Moreover, atrial electromechanical dissociation was associated with a worse prognosis compared to patients in sinus rhythm with mechanical atrial contraction [105]. Consequently, atrial strain analysis showed additive value to identify higher thromboembolic risk patients with cardiac amyloidosis and sinus rhythm compared to conventional clinical workup due to its strength to detect atrial electromechanical dissociation [105]. However, the indication of anticoagulation in this population is still unsettled [106].”
Reference list – page number 16.:
- Vasquez, N.; Ostrander, B.T.; Lu, D.Y.; Ventoulis, I.; Haileselassie, B.; Goyal, S.; Greenland, G.V.; Vakrou, S.; Olgin, J.E.; Abraham, T.P.; Abraham, M.R. Low Left Atrial Strain Is Associated With Adverse Outcomes in Hypertrophic Cardiomyopathy Patients. J Am Soc Echocardiogr 2019, 32, 593-603.e1. doi: 10.1016/j.echo.2019.01.007.
- Kurzawski, J.; Janion-Sadowska, A.; Gackowski, A.; Janion, M.; Zandecki, Ł.; Chrapek, M.; Sadowski, M. Left atrial longitudinal strain in dilated cardiomyopathy patients: is there a discrimination threshold for atrial fibrillation? Int J Cardiovasc Imaging 2019, 35, 319-325. doi: 10.1007/s10554-018-1466-2.
- Cao, S.; Zhou, Q.; Chen, J.L.; Hu, B.; Guo, R.Q. The differences in left atrial function between ischemic and idiopathic dilated cardiomyopathy patients: A two-dimensional speckle tracking imaging study. J Clin Ultrasound 2016, 44, 437-445. doi: 10.1002/jcu.22352.
- Ohtani, K.; Yutani, C.; Nagata, S.; Koretsune, Y.; Hori, M.; Kamada, T. High prevalence of atrial fibrosis in patients with dilated cardiomyopathy. J Am Coll Cardiol 1995, 25, 1162-1169. doi: 10.1016/0735-1097(94)00529-y.
- Bandera, F.; Martone, R.; Chacko, L.; Ganesananthan, S.; Gilbertson, J.A.; Ponticos, M.; Lane, T.; Martinez-Naharro, A.; Whelan, C.; Quarta, C.; Rowczenio, D.; Patel, R.; Razvi, Y.; 106. Lachmann, H.; Wechelakar, A.; Brown, J.; Knight, D.; Moon, J.; Petrie, A.; Cappelli, F.; Guazzi, M.; Potena, L.; Rapezzi, C.; Leone, O.; Hawkins, P.N.; Gillmore, J.D.; Fontana, M. Clinical Importance of Left Atrial Infiltration in Cardiac Transthyretin Amyloidosis. JACC Cardiovasc Imaging 2022, 15, 17-29. doi: 10.1016/j.jcmg.2021.06.022.
- Di Lisi, D.; Di Caccamo, L.; Damerino, G.; Portelli, M.C.; Comparato, F.; Di Stefano, V.; Brighina, F.; Corrado, E.; Galassi, A.R.; Novo, G. Effectiveness and Safety of oral anticoagulants in cardiac amyloidosis: lights and shadows. Current Problems in Cardiology 2022, doi: https://doi.org/10.1016/j.cpcardiol.2022.101188.
Accordingly, the order of references has changed from reference number 101, which is also highlighted in the text.
Once again, we would like to thank the Reviewer for the insightful comments and suggestions! We do believe these resulted in a much improved manuscript which may be acceptable for publication in Diagnostics.
Andrea Agnes Molnar, MD, PhD

Reviewer 2 Report
The manuscript is a review, that summarizes the added value of speckle tracking echocardiography derived left atrial (LA) strain assessment in different clinical scenarios. The manuscript is within the journal’s scope. It is clinically important paper, as it shows real clinical value of evaluation of LA strain in clinical practice.
The manuscript presents a comprehensive analysis of 112 publications and very thorougly and precisely describes methodology of 2DSTE in assessment of LA function, including peculiarities and methodological challenges, LA strain in healthy persons ,as well as in the presence of risk factors, in cardiac diseases (HF, valvular involvement, cancer therapy related cardiac dysfunction, AF and stroke), in athlete’s heart and Covid 19 pneumonia.
The paper is clearly written with two figures of good quality.
The English level of the manuscript is high. The manuscript will be interesting to the wide auditorium of medical specialists, especially cardiologists, imaginers, etc.
I highly recommend the manuscript for publication, owing to clinically relevant topic- consistent summary of LA reservoir function alterations in the clinical setting, that may help to revise the current disease grading and risk scores and may help to detect subtle cardiac dysfunction.
Author Response
Response to Reviewer #2
We would like to thank Reviewer#2 for the encouraging comments and appreciation of our work. as finding it interesting to the wide auditorium of medical specialists. We would like to thank the recommendation for publication owing to the clinically relevant topic, which may help to revise the current disease grading and risk scores and may help to detect subtle cardiac dysfunction.
Andrea Agnes Molnar, MD, PhD
